# A Healthcare Paradigm for Deriving Knowledge Using Online Consumers’ Feedback

**DOI:** 10.3390/healthcare10081592

**Published:** 2022-08-22

**Authors:** Aftab Nawaz, Yawar Abbas, Tahir Ahmad, Noha F. Mahmoud, Atif Rizwan, Nagwan Abdel Samee

**Affiliations:** 1Department of Computer Science, COMSATS University Islamabad, Attock Campus, Attock 43600, Pakistan; 2Department of Rehabilitation Sciences, College of Health and Rehabilitation Sciences, Princess Nourah bint Abdulrahman University, P.O. Box 84428, Riyadh 11671, Saudi Arabia; 3Department of Computer Engineering, Jeju National University, Jejusi 63243, Korea; 4Department of Information Technology, College of Computer and Information Sciences, Princess Nourah bint Abdulrahman University, P.O. Box 84428, Riyadh 11671, Saudi Arabia

**Keywords:** decision-making, home healthcare, healthcare paradigm, pattern recognition, quality measurement, valuable insights

## Abstract

Home healthcare agencies (HHCAs) provide clinical care and rehabilitation services to patients in their own homes. The organization’s rules regulate several connected practitioners, doctors, and licensed skilled nurses. Frequently, it monitors a physician or licensed nurse for the facilities and keeps track of the health histories of all clients. HHCAs’ quality of care is evaluated using Medicare’s star ratings for in-home healthcare agencies. The advent of technology has extensively evolved our living style. Online businesses’ ratings and reviews are the best representatives of organizations’ trust, services, quality, and ethics. Using data mining techniques to analyze HHCAs’ data can help to develop an effective framework for evaluating the finest home healthcare facilities. As a result, we developed an automated predictive framework for obtaining knowledge from patients’ feedback using a combination of statistical and machine learning techniques. HHCAs’ data contain twelve performance characteristics that we are the first to analyze and depict. After adequate pattern recognition, we applied binary and multi-class approaches on similar data with variations in the target class. Four prominent machine learning models were considered: SVM, Decision Tree, Random Forest, and Deep Neural Networks. In the binary class, the Deep Neural Network model presented promising performance with an accuracy of 97.37%. However, in the case of multiple class, the random forest model showed a significant outcome with an accuracy of 91.87%. Additionally, variable significance is derived from investigating each attribute’s importance in predictive model building. The implications of this study can support various stakeholders, including public agencies, quality measurement, healthcare inspectors, and HHCAs, to boost their performance. Thus, the proposed framework is not only useful for putting valuable insights into action, but it can also help with decision-making.

## 1. Introduction

Many people want to remain at home as they get older, especially those with disabilities. However, home healthcare services are becoming increasingly sophisticated and intensive. People with disabilities, chronic diseases, and functional impairments need additional services and support to maintain their independence. When it comes to fulfilling the needs and demands of these populations, home health organizations and other service providers are investigating new models of treatment and payment, as well as the optimum use of their workforce. It is important to consider where home healthcare fits into the wider healthcare system in light of these issues and possibilities. Individuals appreciate the benefits of receiving healthcare at home, and well-managed home healthcare can encourage healthy living and well-being [1].

Home healthcare agencies (HHCAs) are a network of treatment delivered to individuals in their residences by professional staff under the supervision of medical physicians. The Medicare rules are perceived as the standard treatment for all interactions between HHCAs, even if an individual is not insured by Medicare [2]. The star ratings in Medicare are utilized to measure the quality of the service HHCAs provide. According to the Centers for Medicare and Medicaid Services (CMS), over 5.26 million aged and impaired persons were cared for by 10,519 HHCAs throughout the USA in 2019 [3]. HHCAs are the most rapidly rising expenditure in the Medicaid beneficiaries due to the elderly population, a higher number of chronic illnesses, and increased hospital fees [4].

The high quality of services provided by HHCAs is an essential component for patients in order to improve the provided services. In Medicare, star ratings are of significance and are valuable to analyze the performance of quality services regarding the HHCA. It was proposed in July 2015 by CMS and named Quality of Patient Care star ratings (QoPCsr) with a range of 1–5; “1” means an awful experience about service, and “5” means the consumer is delighted about the system [5]. They implemented star ratings as a significant criterion for customers to consider when choosing a home care professional.

The world of HHCAs varies from clinics and other agencies, whereby nurses are employed. For this instance, home healthcare workers operate independently in the field with support services provided by a head office. The nurse–practitioner working partnership has considerably less physician communication, and the surgeon relies on the nurse to render decisions and communicate observations to a larger extent [6]. This high degree of patient control in the home environment and the minimal supervision of informal careers by skilled physicians motivated us to conduct research in HHCAs and uncover the influential features using the star ratings.

Another characteristic of HHCAs is that physicians deliver services in a special environment for everyone. There could be situational factors that pose costs for patients that the health professional cannot remove [7]. Hospitals should have offices for environmental protection to control air pollution, and engineers should guarantee that the staircase height is secure. Home care professionals are unlikely to be trained or have the means to identify and improve patient welfare threats at home. It is essential to determine all the factors for HHCAs with the star ratings to enhance their quality service and increase their revenue as well. Moreover, the influential factors are also helpful in satisfying the customer by providing better services in the Medicare environment.

Influential features are one the best ways to obtain favorable results from unstructured data. The identification of novel features concisely improves the measuring evaluators in the quantitative study. The study of HHCAs enables us to establish an effective framework by utilizing data mining techniques for exploring the best house healthcare facilities, as there is a need to declare an ML model that provides promising outcomes.

Numerous healthcare professionals at home wondered why they have poor star ratings considering comparable facilities and services. The poor satisfaction rate suggests that patients are reduced to home health services, significantly affecting the healthcare provider’s income over time. Home practitioners’ reputation could also be involved in the poor star ratings. Combined with this, it is challenging to continuously examine enormous amounts of data while discovering complex and dynamic characteristics that are likely to occur, but are obscure to humans [8]. The rising digitization of healthcare and the advancement in ubiquitous computing technology has hastened the development of prediction models for deriving knowledge from patient feedback for home care services. The retrieved knowledge could potentially be valuable to various user groups within the healthcare industry, ranging from patients to their respective healthcare practitioners [9].

Therefore, examining Medicare’s star ratings of HHCAs’ data is a need, and the use of artificial intelligence (AI) in gaining insights from these data will help in establishing an effective framework for evaluating the finest home healthcare facilities. Artificial intelligence, known as machine learning, or ML, can be defined as the application of a number of different statistical methods that can be used to produce predictions and decisions based on similarities between what is currently being examined and what is being identified in the past. To fill this gap, we propose an automated ML-based method for gaining insights from the OASIS and Medicaid claim datasets. The utilized approach will yield the influential factors that are highly dependent on the star ratings. A publicly available home healthcare agencies (HHCAs) dataset [10] is utilized to conduct the experimental phase using the different machine learning (ML) techniques. The research aims to gain valuable insights from an unstructured form of HHCAs’ data. The unstructured data are firstly converged into a meaningful form so that machines can easily interpret the data. Then, various statistical techniques are brought into action to find the influential characteristics of the data. In this study, we employ binary and multi-class classification on four renowned machine learning algorithms. In addition, diving deeply into the study, the variable significance for each factor is computed to evaluate each feature’s participation in the HHCAs’ predictive model building. Healthcare-related data are frequently vast and challenging for individuals to swiftly evaluate and interpret. In order to identify and predict different ailments effectively, ML-based models have demonstrated promising outcomes in all medical domains [11]. Analyzing the HHCAs’ data using data mining techniques along with ML approaches can assist in the creation of a framework for accurately identifying the best home healthcare facilities. A blend of statistical along with ML techniques is used to construct an automated predictive framework for extracting knowledge from patient input.

To emphasize the significance of our work, the following are the contributions that this study makes:Binary class and multi-class classification are applied by using four renowned ML models built to justify the robust model for HHCAs’ data.Computing variable significance score for each attribute to analyze the contribution of each indicator in predictive analysis.For experimentation, unstructured data are considered, and statistical techniques are applied to uncover the outperforming indicators.Twelve effective attributes are proposed in this research, which can help in finding the best HHCA. We are the first to explore these features from HHCAs’ data.

The rest of the study is structured as follows: Section 2 covers the literature review, where studies are considered regarding the role of ML in Medicare and the implication of Deep Learning in Medicare. In Section 3, the proposed methodology is discussed along with the proposed framework and a brief description of the data. In Section 4, experiments are performed, and the findings are analyzed in the Results Section. Finally, Section 5 and Section 6 illustrate the conclusion and future work of the study, respectively.

## 2. Related Work

The way doctors and health givers think about disease and treatment has significantly evolved to reflect the changes that have taken place in our patients, our healthcare system, and medical science. The complexity level now present in medicine is beyond the capabilities of the human mind [12]. As a result, the healthcare industry has made substantial use of computer algorithms, which can learn from human decisions [13]. Health information technologies (HITs) have been widely regarded as critical for enhancing the quality of healthcare organizations [14,15,16]. Driven by the significant gains of these technological innovations, whether in clinical or IT fields, the governmental agencies in Europe and America have committed large financial resources to advance HITs’ adoption in the healthcare sector [17,18].

Data analysis tools evaluate various data types and run relevant analyses to gain insights from data records. This is essential when it comes to translating raw patient data into useful information, which can be used to support the decision-making in healthcare organizations [19,20]. Delen [21] proposed a simple classification of analytics that distinguishes three types of analytics: descriptive, predictive, and prescriptive, each defined by the data type and the objective of the study. In the context of predictive analytics, it is possible to predict a particular variable’s future by utilizing probabilistic modeling [22]. With predictive analytics, developers can access flexible and active predictive models for predictions for the future that identify causalities, trends, and hidden correlations between the input and target output. It is not hard to see how predictive analytics can be used in the healthcare industry to help healthcare providers understand the complexities of clinical cost, find the most effective treatment options, and anticipate future healthcare trend lines relying on the habits, lifestyles, and diseases of their patients [23]. Natural language processing (NLP) and Data Mining are mostly applied in predictive-analytics-based approaches [24,25].

Various researchers have studied the impact of important factors using ML techniques in prior studies. Most of the studies provide the solution for binary classification and regression analysis in the medical domain. Some important research to find the influential parameters using the ML methods are discussed here.

The start was to develop and verify a patient registration system for health promotion that allows patients to be classified according to their skilled nursing needs [26]. Nursing theory and experience were combined to create the Community Health Intensity Rating Scale (CHIRS). Groups of public health nursing experts created model definitions for fifteen public health criteria, and they used both patient attributes and essential measures of treatment as descriptive words of nursing care standards. The method was then put through its paces with the help of three home health agency support nurses. A cumulative rating of 560 graphs by two home health organizations was used to validate the system.

Prediction analytics and AI in healthcare face a number of obstacles, including data access, standardization, engagement, computing resource requirements, and the implementation of predictive models [27]. In addition, big data analysis has its own set of issues to deal with [17,28]. It has become increasingly important to explore the use of high-speed cloud computing for both data storage and maintenance, as well as for business intelligence, due to the recent growth of big data in medicine and the advancement of cloud computing [29]. Nevertheless, the analogies between cloud and non-cloud storage and maintenance of massive data are complex. Big data analysis, on the other hand, necessitates high-performance parallel distributed computing algorithms. Data science, bioinformatics, statistical genomics, and other fields have all attempted to address this issue. In addition to the aforementioned medical images and genomic data, these apps also allow for the study of organized data [29,30].

Moreover, rating quality was examined and how confidential rates react to the introduction of a five star rating system for nursing homes [31]. According to their findings, the difference in price among top and bottom facilities grows because of star ratings. The highest-level facility’s prices rose by 4.7 to 6.1% higher than the lowest-level premises’ prices in total. They see more fantastic pricing effects in less mature industries, where buyers can choose from a broader range of nursing homes. The findings show that customers are more receptive to quality reporting where the interface is streamlined and the audiences are less fragmented.

During the literature review, it was noted that there was no similar research on the star rating of reviews using Medicare data with the exception of one paper [32], where the research was conducted on review ratings, but in the context of patient outcomes and not on the influential factors. Therefore, the research being conducted in this paper is entirely novel in that sense. Many research papers identified the influential factors in other areas apart from the review ratings. Conceptually similar papers were selected for the literature review, and the findings are presented in this paper. The association between the Centres for Medicare and Medicaid Services Hospital Star Rating and Patient Outcomes was researched, which is somewhat related to the current study in terms of star ratings and Medicare providers [32]; however, this research is novel in the sense that it focuses just on the home healthcare rating and influential factors driving the rating.

## 3. Proposed Methodology

The methodology for the home healthcare agencies (HHCAs) framework is discussed in this section. Figure 1 supports the precise understanding of the whole approach regarding the methodology and analysis that we adopted to uncover the most influential factors. For this purpose, the HCCAs’ dataset was considered, where four feature selection techniques were employed. First, data pre-processing was carried out on the available data. Second, the double results’ interpretation approach was based on binary and multi-class classification problems utilizing the star ratings as the target variable. Third, four renowned ML models (Random Forest, Deep Neural Network, Support Vector Machine, Decision Tree) were implemented on the HHCA dataset to find the most robust binary and multi-class classification models separately. Moreover, the evaluation of ML models was assessed based on the accuracy, recall, precision, and F-1 score. Finally, the receiver operating characteristic curve (ROC) for each problem (binary, multi-class) is presented to analyze the varied threshold visually for a better understanding of the ML model.

### 3.1. Dataset Description

In this research, we considered the “Home Health Care Agencies (HHCA)” dataset, which is a directory of all Medicare-approved home care services [10]. It consists of “11,176” rows and “70” attributes, including the target variable. The name of the target variable is “Q_p_c_S_r”, which is the ratings of the various customers that have a range of 1–5. In a real dataset, huge attribute names were renamed to shorten them for better understanding. New names for all attributes in the dataset and their old names are depicted in Table 1.

### 3.2. Data Preprocessing

Dataset pre-processing is a data mining method involving transforming raw information into a comprehensible format. Real-life statistics are frequently unreliable, contradictory, and without any patterns or trends, which would certainly include a large number of mistakes [33]. The HHCA dataset’s pre-processing was performed to handle the missing values and remove the inconsistencies from the dataset. A huge number of missing values in variables were handled to make the dataset clean. Only for numeric variables, missing values were imputed by using the mean and median for each respective variable. Moreover, in the HHCA dataset, some columns were in the form of textual information, which also have missing values of more than 70%, which are unable to be filled or predicted. Therefore, we removed the “29” unwanted columns from the dataset, and their names are “P_Name”, “CMS _CN”, “Address”, “ZIP”, “Phone”, “D_Certified”, “F_q_p_c_s_r”, “F_h_p_c_m”, “o_P_S”, “F_o_h_p_c_d”, “F_hh_t_p_r_f”, “F_p_f_d”, “s_c_f_s”, “a_p_v_p_s”, “F_S_R”, “gg_p_f_c”, “w_o_m_a”, “p_o_b”, “p_g_a_b”, “o_p_b_i”, “h_a_an_o”, “b_td_c_m”, “s_h_t_b_a_h”, “c_ER_w”, “ac_c_p_i”, “a_m_i_w”, “Fs_Med”, “D_P_C”, “P_Pe_C”.

Identifying possible outliers is critical, and it is necessary to remove them before performing an analysis [34]. For this purpose, we utilized the Weka software with an “IQR” filter to remove the outliers from the dataset, and “4926” instances out of “11,176” were outliers in the dataset. These were removed from the dataset and made the dataset ready for predictive analysis. Finally, data preparation was performed successfully, and it selected the “41” features along with clean data for the next phase.

### 3.3. Problem Formulation

Two classifications were conducted on the dataset, including multi-class classification [35] and binary classification problems [36]. Star ratings are a multi-class classification problem with a range of 1–5 target classes in the dataset. Therefore, we propose the multi-class classification solution for the problem using the five classes on the dataset by implementing the ML techniques. On the other hand, we divided the star rating into two classes: the “bad” and “good” target class, making it a binary classification problem. On the other hand, we converted the target variable for the binary classification problem by setting the range of 1–3.2 as a bad class and all others as good classes. This division of target variables for binary classification was performed to make the dataset balanced. The unbalanced target class overfits the ML model, which leads to biasness [37]. Therefore, we made a better division for the binary target class to overcome this issue. The performance of the ML techniques was evaluated based on the confusion matrix and critically analyzed by interpreting the results for each model.

Predictive analysis is a significant part of this research for creating good enough and timely decisions using HHCAs using the five machine learning algorithms. In this case study, the HHCA dataset was considered for predictive analysis using feature selection and predictive analysis. We uncovered the most influential features and then built the predictive model using the vital feature. The Weka tool was considered the most powerful predictive analysis for building ML predictive models.

The proposed study utilized ML and DL models to show the effectiveness of the proposed features. SVM is a powerful algorithm due to its kernel techniques such as the linear, polynomial, and RBF kernel. These kernel tricks are used to transform the data from one feature pace to another feature space, where the data are more linearly separable [38]. Along with SVM, RF and DT were selected from the ML techniques for binary and multi-class classification problems. Moreover, from deep learning models, the DNN was selected, and the results were compared with other ML models. DNN performed well in binary classification compared to the other models.

## 4. Results

A deep analysis is conducted in this portion of the research. All the steps and outcomes from the previous methodology section are employed here to gain valuable insights from the HHCA dataset. Briefly, experiments were conducted on the available data by employing statistical techniques to uncover the helpful attributes from the available data. Moreover, a concise approach was performed using machine learning to build sophisticated models to obtain valuable outcomes from those outperformed characteristics.

### 4.1. Feature Subset Selection

This subsection performs the feature subset selection for HHCAs using the “40” features to find the most influential indicators. To fulfil this need, we considered the various filters of the WEKA tool for feature selection and dimension reduction. The ranker method and greedy search algorithms find the most important feature from the dataset [39,40]. Principal component analysis (PCA) was used to reduce the dimension of the dataset before applying the predictive model to obtain the best score [41]. Finally, the HHCA dataset was normalized before feature engineering to make it within ranges for better results.

#### 4.1.1. Correlation Score with Ranker Method

The correlation score of each feature was determined with the help of the Ranker method in Weka [42]. The value for the correlation was 1 to −1. A more positive correlation means that the variables are highly correlated with the target class, and a negative correlation means they have an imperfect correlation with each other. If the correlation value is 0, then there is no correlation between the variables. Moreover, sometimes, the correlation values became negative, which means that the variables are negatively correlated, which is very bad. In the table on using Weka, the correlation score of each feature is shown concerning the target variable. This method chooses the important variables highlighted in Table 2 because the correlation value is 0 and goes towards negative after that. Therefore, we left these variables because they were not helpful.

#### 4.1.2. CFS Subset Eval Using Greedy Stepwise

The forward selection of greedy stepwise was considered here for the subset evaluation of the variable’s selection. This approach is also helpful in uncovering the most influential variables from the dataset [43]. The ranking with CFS is presented in Table 3. Using this approach, a total of “12” features were selected, as listed above. This analysis only selected the features with the help of using the forward selection.

#### 4.1.3. Relief F Attribute Eval with Ranker

This method also shows the correlation score of the features, and the results are given in Table 4. This filter chooses the total number of “37” features out of “40”, as shown in Table 4. The correlation score for each feature is also listed to explain its importance for the “Q_p_c_s_r” target class.

#### 4.1.4. Using the PCA Selection

Principal component analysis (PCA) was beneficial to deal with the dimensionality reduction of the dataset [41]. It is a dimension-reducing technique utilized to minimize the size of large volumes of data by converting a considerable number of variables into yet another smaller one, which also retains most details in the more extensive collection [44]. We utilized PCA for the HHCA dataset, and its results are shown in Table 5.

The total, “27” dimensions were used here, depending on the threshold values. Here, we set the default value confidence interval up to 95%, which is the most appropriate for the analysis. We also set the threshold value following our own needs.

### 4.2. Selecting Feature Selection Algorithm

In this case study, “4” feature selection algorithms were applied in the HHCA dataset to find the best features among them and remove the unwanted or uncorrelated features. Each of the algorithms works to remove several features using their threshold values. Now, selecting one feature is important for approaching further. After applying the feature selection algorithms, the datasets were saved for comparing the capabilities of the different techniques. Therefore, after applying four feature evaluation techniques, we obtained four datasets: Correlation Attribute Evaluation Dataset (CAED), Principal Component Analysis Dataset (PCAD), Relief-F Attribute Evaluation Dataset (RAED), and CFS Subset Evaluation Dataset (CSED). We selected Random Forest (RF) as the baseline classifier to check the capabilities. The performances were checked for binary classification and multi-class classification. A broad discussion on binary and multi-class classification is given in the later sections. Table 6 shows the RF classification results for binary classification and Table 7 shows them for multi-class classification.

Here, a clear winner is the CFS subset evaluation dataset, where the feature was reduced from 40 to 12, a massive selection, yet scoring best among the other selection criteria. Relief F feature selection performed rather close, but in terms of reducing the features, CFS decreased to 28 features, whereas Relief F reduced to only 3. Table 8 shows the time taken by both datasets, and here, there was almost a 50% time complexity reduction for CFS subset evaluation. Therefore, there was a huge time gap, and the CFS subset evaluation also increased the model accuracy. Further binary and multi-class classification processing was performed on the CFS subset evaluation dataset.

### 4.3. HHCA Predictive Analysis

In this subsection, four ML models, including the Deep Neural Network (DNN), Random Forest (RF), Support Vector Machine (SVM), and Decision Tree (DT), were implemented on the HHCA dataset. Two types of classification models (binary and multi-class) were implemented to build an effective framework for HHCAs. For this purpose, Weka tools were utilized to make the predictive models for binary and multi-class classification problems. Both approaches have their significance due to their pros and cons. The main objective of the predictive analysis is to find the most robust classifier for the HHCAs that gives the best results. The outperforming ML models for binary and multi-class classification problems were assessed using the four evaluation metrics (accuracy, precision, recall, and F-1 score). The experimental details of both techniques are presented below.

#### 4.3.1. Binary Classification

This part of the research was based on the binary classification data of HHCAs. The data were labeled with only two classes, “Good” and “Bad”. As we employed only two classes of data to build the ML model, this is why the part was declared as a “binary classification” approach. Various ML models were tested and trained using 10-fold cross-validation. We used renowned models to depict high accuracy, precision, recall, and F-1 measure results. Table 9 below is clear evidence of the results we recorded using the Weka tool.

Table 9 illustrates that the Decision Tree gave low results in terms of the evaluation metrics, where the accuracy, precision, recall, and F-1 measure were 94.3%, 94.3%, 94.3%, and 94.3%, respectively. However, the Deep Neural Network model gave the best results covering the same metrics and having values of 97.4%, 97.4%, 97.4%, and 97.4%, respectively. Thus, the Deep Neural Network outperformed all other applied models. Moreover, in terms of HHCA binary class data, we were the first to explore the results using the Deep Neural Network model, giving valid results compared with all other applied models.

#### 4.3.2. Multi-Class Classification

This section of the paper is based on the multi-class classification data of HHCAs. The data were labeled with five different classes, which were “Best”, “Good”, “Average”, “Bad”, and “Very Bad”. As we utilized only multiple classes (five different classes) of data for building the ML model, the part was declared as a “multi-class classification” approach. Various ML models were tested and trained using 10-fold cross-validation. We used renowned models to depict the high accuracy, precision, recall, and F-1 measure results. Table 10 below is clear evidence of the effects we recorded using the Weka tool.

Table 10 presents that the Decision Tree gave low results in terms of the evaluation metrics, where the accuracy, precision, recall, and F-1 measure were 86.7%, 86.7%, 86.7%, and 86.7%, respectively. In contrast, the Random Forest model gave the best results covering the same metrics and had 91.9%, 91.8%, 91.9%, and 91.7%, respectively. Thus, we considered that the Random Forest outperformed all other applied models in the comparison.

### 4.4. Comparative Analysis Using the ROC

In this case, we evaluated the ROC results for binary and multi-class classification. As we know, the binary class records have the labels “Good” and “Bad”; therefore, the outcome of the classifier is shown in Figure 2. We know that the Deep Neural Network gave outperforming binary class results (as discussed in Table 9). Therefore, we used the Deep Neural Network to obtain the results of the outcome. We employed the same technique of 10-fold cross-validation here using Weka.

Figure 2 is clear evidence for obtaining the ROC results of binary class classification using Deep Neural Networks. The ROC results for the “Good” class were 99.65%. However, the outcome for the “Bad” class of the HHCAs’ data was 99.68%. Therefore, both graphs are separately depicted in Figure 2 and Figure 3.

There are five different classes in the multi-class approach, and we evaluated five different ROCs for each class, illustrated in Figure 3. As Table 10 describes, the Random Forest model was the best among all multi-class classification models; therefore, we used the same model. Ten-fold cross-validation was used to achieve the results.

Figure 3 presents that each class had different ROC results. We evaluated that “Best”, “Good”, “Average”, “Bad”, and “Very Bad” has ROC values 99.61%, 98.83%, 98.70%, 99.18%, and 99.70%, respectively.

## 5. Discussion and Conclusions

This research proposed a solution for the Medicare industry to uncover the most influential attributes using the star ratings. Two types of ML techniques were implemented, which included binary and multi-class classification. The HHCA dataset was utilized here, which contains 70 features along with the target variable (star rating). The data pre-processing was performed along with handling missing values, removal of inconsistencies, and elimination of outliers from the dataset. Afterward, feature engineering was conducted using Weka, and four different attribute selection filters (CAE, RA, PCA, CSE) were applied to locate the most impactful attributes for binary and multi-class classification problems. The RF model was chosen for these “4” filters to find the best feature selection technique, and the findings depicted that the CSE filter had the best performance for both techniques (binary, multi-class) using the “12” features. The name of the important factors are O_H_H_A_Ser, h_t_p_c_t_m, r_c_f_s, f_c_t_p_f_c, b_m_a, h_g_o_b, g_ b_a_b, p_b_i, i_h_a_o, a_d_c_b_m, t_b_a_t_h, and P_iS_a. After selecting the best from the 70 available attributes, four renowned machine learning models (DT, SVM, DNN, RF) were utilized for binary and multi-class classification using the “12” features. The ML model’s performance was analyzed based on four evaluation metrics: accuracy, precision, recall, F-1 score. For better results, the hyperparameters’ tunings were also considered, and we tested the models with the best hyperparameters. The findings showed that the DNN and RF models outperformed and achieved the highest score among all other models for binary and multi-class classification, respectively. The ROC was also a significant evaluation metric for finding the significance of the model. We computed the ROC results for each class in both binary and multi-class classification. The findings of this research are helpful in the healthcare domain to improve the customer experience using the influential features for better results.

The twelve significant extracted features affecting Medicare’s star rating had the following features: Offers Home Health Aide Services, how often the home health team began their patients’ care in a timely manner, how often the home health team determined whether patients received a flu shot for the current flu season, How frequently did the home health care staff follow the doctor’s recommendations, provide foot care, and provide education to people with diabetes, Frequency with which patients improved in their capacity to walk and move, Rate with which patients improved their ability to get out and back into bed, How frequently patients improved their ability to shower, Note on the frequency with which patients’ breath improves, Rate at which surgical incisions healed or improved, Rates at which patients improved their oral medication adherence, Frequency with which home health care recipients were hospital admission, PPR Risk-Standardized Rate (Upper Limit).

Initiation of Care at the Appropriate Time was one of the important features, h_t_p_c_t_m, that was yielded in this study, and it affects the Medicare star rating. It shows the percentage of home health quality occurrences when treatment began or resumed. Timely Initiation of Care [45] came first and the feature “Offers Home Health Aide Services”, O_H_H_A_Ser. This is a vital feature especially for those with disabilities. However, further analysis is needed to improve this service as Medicare does not cover home health personal care aides as a stand-alone service. It only pays for a home health personal care aide when an individual also receives skilled nursing care or rehabilitation services through home health.

Appropriate timing is one of the important aspects preventing costly rehospitalizations and improving patient outcomes. When home healthcare was delayed after hospital discharge, the patients were more likely to experience a 30-day rehospitalization, and the association between it and the rehospitalization of diabetes patients was investigated in [46], while influenza immunization received for the current flu season is another important feature, r_c_f_s, considering that influenza has been linked to 12,000 to 56,000 deaths in the United States alone each year and that geriatric adults, those ≥65 years old, are the most vulnerable to severe infection and account for up to 85% of these deaths [47].

Improvement in bathing, g_b_a_b, was another feature that was yielded in this investigation that affects the star rating. The percentage of home health quality episodes in which the patient improved his/her ability to bathe himself/herself on his/her own is reported by the “Improvement in Bathing [45]” feature. In [48], the effect of this feature was investigated to understand how alterations in the physical capacity of an older adult affect his/her preferences in bathing, as well as how the care environment incorporates these alterations.

Another aspect of this study’s findings that impacts the star rating was the “Improvement in the status of surgical wounds”, i_h_a_o. Home care wound management was tested in [49] to see if oral antibiotics and the wound and patient variables affected the efficacy and surgical site infection rate compared to hospital-based wound management.

## 6. Future Work

The future work of this research is a feasible guideline for researchers. The data specialist can bring valuable insights from the available data residing on the website (cms.gov accessed on: 16 March 2022). In the dataset, a few columns are text-based, which we did not consider for this research. These could be considered for future analysis. Moreover, for future studies, some of the columns were removed because of the many missing values. These missing values can be handled by employing regression techniques for building the solution (Meeyai, 2016). Thus, significant features can be discovered by considering the neglected parameters. Modern machine learning models are very sophisticated and favorable to adopt by the agencies. Thus, deep learning and ensemble models can be helpful to optimize the performance of models to obtain the optimal results.

## Figures and Tables

**Figure 1 healthcare-10-01592-f001:**
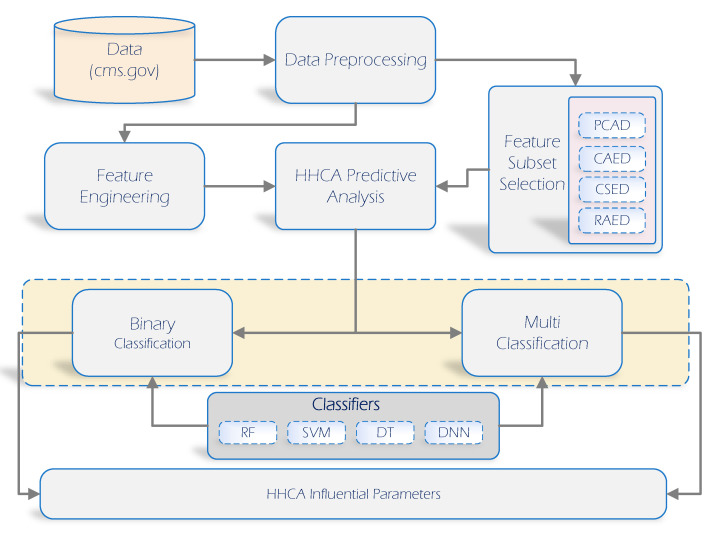
HHCA methodology and analysis overview.

**Figure 2 healthcare-10-01592-f002:**
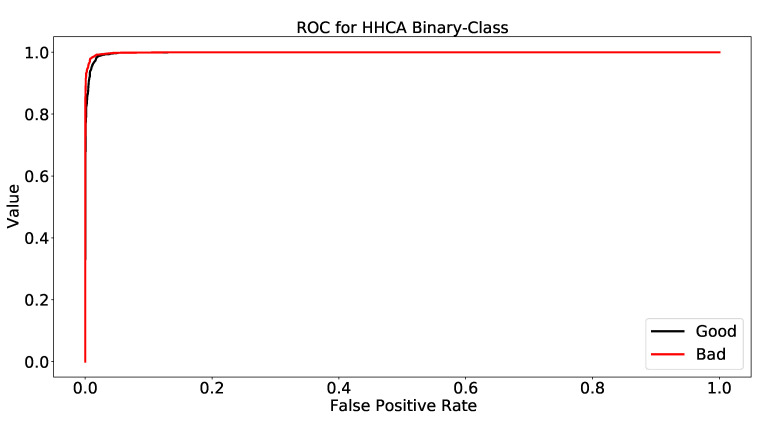
ROC for HHCA binary class.

**Figure 3 healthcare-10-01592-f003:**
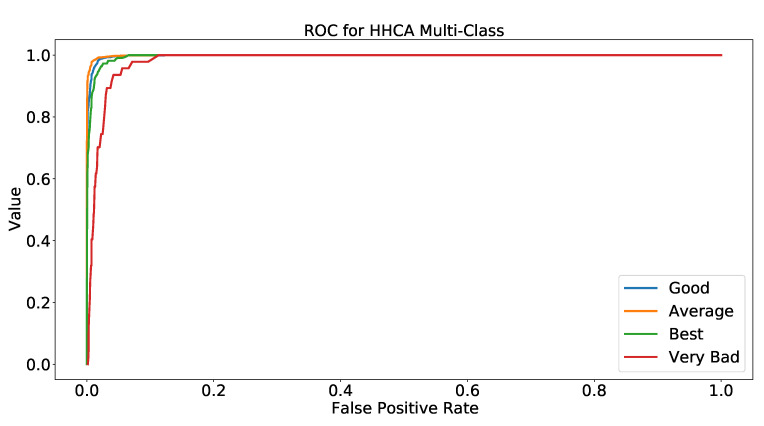
ROC for HHCA multi-class classification.

**Table 1 healthcare-10-01592-t001:** HHCA dataset attributes’ description.

New Names of Attributes	Old Names for Attributes
State	Name of state
CMS _CN	Number of CMS certification
P_Name	Name of providers’
Address	Details of address
City	Name of city
ZIP	ZIP code
Phone	Details of phone number
T_Ownership	Ownerships’ detail
O_N_ C_Ser	Provides services in the field of nursing
O_Phy_ T_Ser	Provides services for physical therapy
O_Occ_The_Ser	Provides services in occupational therapy
O_Sp_Pa_Ser	Provides services in speech pathology
O_Medi_S_Ser	Provides services in medical social
O_H_H_A_Ser	Provides services in home aide health
D_Certified	Details of Certification date
F_q_p_c_s_r	Notes on the star rating for the quality of patient treatment
h_t_p_c_t_m	Rates at which home health care services were initiated in a timely way for their patients
F_h_p_c_m	The rate of on-time patient care starts by the home health staff is noted
h_t_p_d	When individuals (or associated families/careers) were informed regarding their medications by the home health care provider
F_o_h_p_c_d	A footnoted list of how frequently home health workers provided drug information to patients (or family members)
H_t_p_r_f	How frequently the household health staff examined clients’ chances of falling
F_hh_t_p_r_f	Whenever the home health care provider noticed a patient was at danger of falling, they would do a checkup
h_t_c_p_d	Frequency of home care providers’ checks regarding depression
F_p_f_d	Note on how frequently the home health care staff checks patients regarding depression
r_c_f_s	Rate at which home health care workers checked to see whether their patients were getting a flu vaccination this year
s_c_f_s	Note on how frequently home health workers checked to see whether patients were getting a flu vaccination this year
p_v_p_s	Pneumococcal vaccination frequency as monitored by home health care providers (shot of pneumonia)
a_p_v_p_s	Pneumococcal vaccination frequency note for patients cared for by the home health care staff (shot of pneumonia)
f_c_t_p_f_c	How frequently did the home health care staff follow the doctor’s recommendations, provide foot care, and provide education to people with diabetes
gg_p_f_c	Note on the frequency with which home health aides followed doctors’ directions to treat patients’ feet and instruct them on how to better take care of them
b_m_a	Frequency with which patients improved in their capacity to walk and move
w_o_m_a	This footnote refers to the frequency with which patients improved in their ability to walk or move about
h_g_o_b	Rate with which patients improved their ability to get out and back into bed
p_o_b	Note on the frequency with which patients improved their ability to just get out and back into bed
g_ b_a_b	How frequently patients improved their ability to shower
p_g_a_b	Note on the frequency which patients improved their ability to shower
p_b_i	Rate with which patients experienced an improvement in their breathing
o_p_b_i	Note on the frequency with which patients’ breath improves
i_h_a_o	Rate at which surgical incisions healed or improved
h_a_an_o	Note on the frequency with which patients’ surgical wounds healed or cured
a_d_c_b_m	Rates at which patients improved their oral medication adherence
b_td_c_m	Note indicating the frequency with which individuals improved their oral medication adherence
t_b_a_t_h	Frequency with which home health care recipients were hospital admission
s_h_t_b_a_h	Note indicating the frequency with which home health care recipients were hospital admission
E_ w_b_a	Frequency with which home health care recipients need unscheduled, emergent treatment in the emergency room without being hospitalized
c_ER_w	Note indicating the frequency with which home health care recipients need unscheduled, emergent treatment in the emergency room without being hospitalized
p_u_i	Skin integrity alters after hospitalization due to pressure ulcers or injuries
ac_c_p_i	Note for skin integrity alters after hospitalization due to pressure ulcers or injuries
m_i_w_c_t	How frequently medication problems were resolved immediately after doctors gave their advice
a_m_i_w	Note indicating the frequency for medication problems which are resolved immediately after doctors gave their advice
D_Num	Numerator for DTC
DT_D	Denominator for DTC
D_O_R	Observation rate for DTC
D_S_R	Risk standardized rate for DTC
D_L_L	Lower limit of risk standardized rate for DTC
D_U_L	Upper limit of risk standardized rate for DTC
D_P_C	Categorization’s performance for DTC
F_S_R	Note of risk standardized rate for DTC
P_Nume	Numerator for PPR
P_Dor	Denominator for PPR
P_R_O_R	Observation rate for PPR
P_RS	Risk standardized rate for PPR
PS_R_L	Lower limit of risk standardized rate for PPR
P_iS_a	Upper limit of risk standardized rate for PPR
P_Pe_C	Categorization’s performance for PPR
Fo_P_St	Note of risk standardized rate for PPR
H_c_na	Cost per episode of treatment for Medicare at this facility, versus the national average for Medicare expenditures
Fs_Med	Note for the cost per episode of treatment for Medicare at this facility, versus the national average for Medicare expenditures
No_p_epi	Count of episodes used to determine company’s per-episode Medicaid expenditure relative to all organizations (National)
Q_p_c_s_r	Quality of patient care star rating (Target/Label Class)

**Table 2 healthcare-10-01592-t002:** Correlation score of each feature.

Method		Correlation Ranking Filter
Ranking	Correlation Score	Feature Name
1	0.8473	g_ b_a_b
2	0.82911	b_m_a
3	0.79995	a_d_c_b_m
4	0.7426	h_g_o_b
5	0.7411	p_b_i
6	0.40886	h_t_p_c_t_m
7	0.27763	r_c_f_s
8	0.26141	i_h_a_o
9	0.23979	D_L_L
10	0.21457	No_p_epi
11	0.20649	p_v_p_s
12	0.19951	D_S_R
13	0.19759	h_t_p_d
14	0.19353	f_c_t_p_f_c
15	0.16222	D_O_R
16	0.14267	D_U_L
17	0.13985	DT_D
18	0.13911	m_i_w_c_t
19	0.12408	H_t_p_r_f
20	0.1028	h_t_c_p_d
21	0.10239	H_c_na
22	0.09768	O_Medi_S_Ser
23	0.08679	D_Num
24	0.07364	P_Dor
25	0.06233	O_Phy_ T_Ser
26	0.06005	O_Occ_The_Ser
27	0.05155	O_Sp_Pa_Ser
28	0.03039	City_c
29	0	O_N_ C_Ser
30	−0.00119	P_Nume
31	−0.00363	E_ w_b_a
32	−0.0217	T_Ownership_c
33	−0.02291	O_H_H_A_Ser
34	−0.0335	PS_R_L
35	−0.07793	P_R_O_R
36	−0.09955	P_RS
37	−0.09985	State_c
38	−0.12648	t_b_a_t_h
39	−0.14147	P_iS_a
40	−0.15858	p_u_i

**Table 3 healthcare-10-01592-t003:** Ranking with CFS subset Eval filter.

Ranking	Feature Name
1	O_H_H_A_Ser
2	h_t_p_c_t_m
3	r_c_f_s
4	f_c_t_p_f_c
5	b_m_a
6	h_g_o_b
7	g_ b_a_b
8	p_b_i
9	i_h_a_o
10	a_d_c_b_m
11	t_b_a_t_h
12	P_iS_a

**Table 4 healthcare-10-01592-t004:** Ranking with filter ReliefFAttributeEval.

Ranking	Score	Feature Name
1	0.0583679	a_d_c_b_m
2	0.0574577	g_ b_a_b
3	0.0568582	b_m_a
4	0.0425037	p_b_i
5	0.0393541	h_g_o_b
6	0.0377682	r_c_f_s
7	0.0362624	p_v_p_s
8	0.0336944	t_b_a_t_h
9	0.0318255	E_ w_b_a
10	0.0223106	P_iS_a
11	0.0210561	City_c
12	0.0200106	i_h_a_o
13	0.0198618	h_t_p_c_t_m
14	0.0192887	D_O_R
15	0.0188932	D_U_L
16	0.0183911	State_c
17	0.0181756	m_i_w_c_t
18	0.0173318	H_c_na
19	0.016835	P_R_O_R
20	0.0168096	D_L_L
21	0.0162158	P_RS
22	0.016175	PS_R_L
23	0.0160877	D_S_R
24	0.0158999	f_c_t_p_f_c
25	0.0117701	p_u_i
26	0.0088474	T_Ownership_c
27	0.0082658	h_t_c_p_d
28	0.0071862	h_t_p_d
29	0.0056432	No_p_epi
30	0.0047277	H_t_p_r_f
31	0.0031908	D_Num
32	0.0023015	P_Nume
33	0.0015753	DT_D
34	0.0014135	O_Medi_S_Ser
35	0.0008787	P_Dor
36	0.0006544	O_H_H_A_Ser
37	0.0000953	O_Sp_Pa_Ser
38	0	O_N_ C_Ser
39	−0.0001223	O_Occ_The_Ser
40	−0.0011722	O_Phy_ T_Ser

**Table 5 healthcare-10-01592-t005:** Feature dimensions with PCA.

Ranked Score	Ranks	Feature’s Dimensions
0.8484	1	−0.273D_L_L-0.266No_p_epi-0.262DT_D-0.248b_m_a-0.246g_ b_a_b…
0.741	2	−0.408P_Nume-0.371D_Num-0.37P_Dor-0.34DT_D-0.293PS_R_L…
0.655	3	−0.439D_U_L-0.425D_O_R-0.423D_S_R-0.369D_L_L+0.198a_d_c_b_m…
0.5821	4	−0.466P_RS-0.419P_iS_a-0.387P_R_O_R-0.379PS_R_L-0.21g_ b_a_b…
0.5146	5	0.48 O_Occ_The_Ser+0.457O_Sp_Pa_Ser+0.441O_Phy_T_Ser+0.36 O_Medi_S_Ser+0.191P_RS…
0.4659	6	−0.38h_t_c_p_d-0.377h_t_p_d-0.34H_t_p_r_f-0.338m_i_w_c_t-0.257p_v_p_s…
0.4301	7	−0.607p_v_p_s-0.545r_c_f_s+0.247H_t_p_r_f+0.231f_c_t_p_f_c+0.221h_t_c_p_d…
0.4002	8	0.467E_ w_b_a+0.433State_c-0.405O_H_H_A_Ser+0.36 T_Ownership_c-0.248r_c_f_s…
0.3709	9	0.688t_b_a_t_h+0.596H_c_na-0.158E_w_b_a-0.155h_t_c_p_d-0.129State_c…
0.3442	10	0.753City_c+0.417p_u_i-0.293State_c+0.282E_ w_b_a-0.17f_c_t_p_f_c…
0.3183	11	−0.7O_H_H_A_Ser+0.388City_c-0.366E_w_b_a-0.361p_u_i+0.171f_c_t_p_f_c…
0.2933	12	0.603p_u_i-0.372E_w_b_a-0.264f_c_t_p_f_c-0.263O_H_H_A_Ser-0.26T_Ownership_c…
0.2689	13	0.807T_Ownership_c-0.413i_h_a_o-0.22E_w_b_a+0.139h_t_c_p_d-0.126State_c…
0.245	14	0.615i_h_a_o-0.341E_w_b_a+0.34 m_i_w_c_t+0.315p_u_i+0.233T_Ownership_c…
0.2227	15	0.389f_c_t_p_f_c+0.355h_t_p_c_t_m-0.349i_h_a_o-0.34H_t_p_r_f-0.325t_b_a_t_h…
0.2013	16	−0.549State_c+0.385i_h_a_o-0.371City_c-0.344m_i_w_c_t-0.226O_H_H_A_Ser…
0.1809	17	−0.605f_c_t_p_f_c-0.413t_b_a_t_h-0.363E_w_b_a+0.328m_i_w_c_t-0.238p_u_i…
0.1609	18	−0.467State_c+0.38m_i_w_c_t-0.294f_c_t_p_f_c+0.288h_t_p_d-0.268h_t_p_c_t_m…
0.1422	19	0.781h_t_p_c_t_m-0.457H_c_na+0.246t_b_a_t_h-0.208f_c_t_p_f_c-0.145p_u_i…
0.1255	20	0.512H_t_p_r_f+0.43O_Medi_S_Ser-0.423h_t_p_d-0.355h_t_c_p_d+0.287m_i_w_c_t…
0.1097	21	−0.657O_Medi_S_Ser-0.424h_t_p_d+0.296O_Phy_T_Ser+0.28 m_i_w_c_t+0.235H_t_p_r_f…
0.0946	22	−0.609h_t_c_p_d+0.54h_t_p_d+0.379H_t_p_r_f-0.298m_i_w_c_t-0.204O_Medi_S_Ser…
0.0814	23	−0.746P_R_O_R+0.437P_iS_a-0.253p_b_i+0.212P_RS+0.163D_O_R…
0.0709	24	−0.757h_g_o_b+0.53a_d_c_b_m+0.186g_b_a_b-0.126p_b_i+0.116O_Sp_Pa_Ser…
0.0604	25	0.691p_b_i+0.457O_Sp_Pa_Ser-0.307O_Phy_T_Ser-0.235b_m_a-0.184O_Medi_S_Ser…
0.0503	26	0.61O_Sp_Pa_Ser-0.464p_b_i-0.423O_Phy_T_Ser+0.319h_g_o_b-0.216O_Medi_S_Ser…
0.0421	27	0.659No_p_epi-0.412P_Nume+0.21 P_iS_a-0.209D_U_L-0.197p_v_p_s…

**Table 6 healthcare-10-01592-t006:** Performance of RF (binary classification).

Dataset	Accuracy	F-Measure	AUROC
CAED	95.77 ± 0.96	95.8 ± 0.71	91.1 ± 1.28
RAED	96.24 ± 1.09	96.2 ± 0.82	99.5 ± 0.24
PCAD	93.90 ± 1.63	93.9 ± 1.47	98.8 ± 0.47
CSED	96.97 ± 0.76	97.0 ± 1.06	99.7 ± 0.08

**Table 7 healthcare-10-01592-t007:** Performance of RF (multi-class classification).

Dataset	Accuracy	F-Measure	AUROC
CAED	88.67 ± 1.28	88.5 ± 0.46	97.0 ± 1.01
RAED	90.22 ± 1.07	89.9 ± 1.07	98.4 ± 0.75
PCAD	83.97 ± 2.41	82.9 ± 0.91	96.1 ± 0.85
CSED	91.80 ± 0.76	91.7 ± 1.12	98.8 ± 0.47

**Table 8 healthcare-10-01592-t008:** Time comparison of different feature combinations.

Dataset	Time Taken
Binary Classification	
CSED	1.2
RAED	2.08
Multi-class Classification	
CSED	1.47
RAED	2.16

**Table 9 healthcare-10-01592-t009:** Models’ performance for binary classification.

Model Name	Accuracy	Precision	Recall	F-1 Score
SVM	97.0 ± 0.74	97.1 ± 1.24	97.1 ± 1.20	97.1 ± 1.27
DT	94.3 ± 0.74	94.3 ± 0.81	94.3 ± 0.71	94.3 ± 1.49
RF	97.0 ± 0.47	97.02 ± 0.91	97.0 ± 0.89	97.0 ± 0.73
DNN	97.4 ± 0.39	97.4 ± 0.27	97.4 ± 0.63	97.4 ± 0.92

**Table 10 healthcare-10-01592-t010:** Models’ performance for multi-class classification.

Model Name	Accuracy	Precision	Recall	F-1 Score
SVM	89.7 ± 1.86	89.5 ± 1.18	89.7 ± 1.01	89.3 ± 0.98
DT	86.7 ± 1.43	86.7 ± 2.36	86.7 ± 1.55	86.7 ± 1.79
RF	91.9 ± 0.33	91.8 ± 0.27	91.9 ± 0.64	91.7 ± 0.31
DNN	88.1 ± 2.07	87.4 ± 1.96	88.1 ± 1.73	86.9 ± 2.32

## Data Availability

We declare that the data considered for this research are original and were collected by the authors for gaining insights. Moreover, the data mining and ML tools considered for this research are freely available, and we built the models in accordance with our own scenario.

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
