# Peer review of "A Healthcare Paradigm for Deriving Knowledge Using Online Consumers’ Feedback"

_healthcare, 2022, doi:10.3390/healthcare10081592_

Round 1

Reviewer 1 Report

This is an excellent project.  Much needed and timely. The English translation needs to be carefully editied,  Word use seems to be fron thesaurus and someties simply inaccurate and not  understandable.  Many of the references are quite outdated and should be reviewed.,

But overall great project.  The conclusions could be improved and made more useful to the averge reaader,

Author Response

Dear Reviewer,

We have updated our manuscript as per your and other reviewers comments, kindly revise the updated version of the manuscript.

Regards

Reviewer 2 Report

Comments on Manuscript encoded: healthcare-1836503

This study examined how much the healthcare attributes take part in finding the best home healthcare agencies, identified a list of attributes reside that are significant and strongly assist in building the best home healthcare agencies (HHCA) framework, and determined which Machine Learning model is more effective to experience substantial performance such that a robust model can be built for the best HHCA with the purpose of assisting various stakeholders, including public agencies, quality measurement, healthcare inspectors, and HHCAs, to boost their performance and help them with the decision-making processes. The study is very well-written, and the overall presentation is engaging.

There is a great deal to appreciate about this study. A reasonably solid rationale, on both conceptual and empirical bases, is developed for a study with potential impact on home healthcare services. Theories and pre-existing frameworks that informed the research question are described up front. The data collection and analyses are clear and detailed enough for readers to gain a sense of how the analysis of the phenomena evolved. The findings are aligned with the principles of the methodology used.

The authors have my congratulations on this account. I would only ask the authors to revise the manuscript for conciseness and brevity.

Author Response

(The authors gave the same response as above.)

Reviewer 3 Report

The author's purpose A Healthcare Paradigm for Deriving Knowledge Using Online Consumers’ Feedback. The contributions of this study is good and clear. Overall, the paper is an interesting one with reasonable solutions and promising experimental results. However, there are quite a few issues to be resolved before considered for publications.

1.      Section 1, Introduction section: The introduction needs to explain more clearly the aims of the paper, and the exact way that it could be assistive to the related research community for healthcare. Explain why machine learning is important for healthcare applications for this cite this work which is related to your work https://doi.org/10.1155/2022/2550120 .

2.      As it is a research paper remove the research questions in the introduction part, as it is not a thesis.

3.      In addition, I feel that there are seven contributions of your study which are too much. There is no need of first three contributions. Please also point out the novelty of the paper too with major contributions.

4.      Section 2, Related Work: The paper also needs to present additional related works and methods. In addition, it needs to explain in better detail for each paper, the methodology of it as well as the findings. Please explain how it builds on and extends previously conducted studies in the literature.

5.      Add some recent papers related to machine learning on various diseases

https://doi.org/10.1109/ACCESS.2022.3174599

6.      - Section 3 needs to present in better detail core aspects of the author's work of your proposed models. Explain the purpose of your proposed machine learning models of SVM, Decision Tree, Random Forest, and Deep Neural Networks.

7.      Explain how you extract the features, the table 1 is useless here regarding features description.

8.      The actual architecture need to be presented in better detail and the layers need to be substantiated

9.      Section 4 needs to present in better detail the aims of the experimental study and the results.

10.  The figure 2 and figure 3 ROC-AUC curves values should mention rather good , average, best.

11.  - There should be a discussion section in which compared with previous state-of – the art- work.

12.  - The results need a deeper and more complete discussion and also the possible connection with existing works and results in the literature.

Author Response

(The authors gave the same response as above.)
